# A Green Water-Soluble Cyclophosphazene as a Flame Retardant Finish for Textiles

**DOI:** 10.3390/molecules24173100

**Published:** 2019-08-26

**Authors:** Thomas Mayer-Gall, Dennis Plohl, Leonie Derksen, Dana Lauer, Pia Neldner, Wael Ali, Sabine Fuchs, Jochen S. Gutmann, Klaus Opwis

**Affiliations:** 1Deutsches Textilforschungszentrum Nord-West gGmbH, Adlerstr. 1, D-47798 Krefeld, Germany; 2University Duisburg-Essen, Institute of Physical Chemistry and Center for Nanointegration, Duisburg-Essen, Universitätsstraße 2, D-45117 Essen, Germany; 3Hochschule Hamm-Lippstadt—University of Applied Sciences, Marker Allee 76-78, D-59063 Hamm, Germany

**Keywords:** phosphazene, flame retardant, textile, cotton, PET, PA, blend, TGA, MCC, fire testing

## Abstract

Poly- and cyclophosphazenes are excellent flame retardants but currently, are not used as textile finishing agents because water-soluble and permanent washing systems are missing. Here, we demonstrate for the first time, the successful usage of a water-soluble cyclotriphosphazene derivative for textile finishing for cotton, different cotton/polyester, and cotton/polyamide blend fabrics. A durable finish was achieved using a photoinduced grafting reaction. The flame retardant properties of the various fabrics were improved with a higher limiting oxygen index, a reduced heat release rate, and an exhibition of intumescent. Furthermore, the finished textiles passed several standardized flammability tests.

## 1. Introduction

Cotton (CO), polyester (PET), and polyamide (PA) based textiles are omnipresent and irreplaceable in our day-to-day life. Typical applications are curtains, carpets, bedding, or upholstered furniture. These materials represent a potential hazard due to their high flammability [1]. Flame retardant textiles are produced by polymer blending or finishing with inorganic salts, such as aluminum or magnesium hydroxide, organohalogens, for example, chlorinated paraffins, bromobiphenyl ethers or bromobisphenols, or formaldehyde-based flame retardants [1,2,3,4]. Due to their toxic potential, REACH has banned all low molecular weight halogenated flame retardants. Exceptional cases are given for special applications, for example, aircraft and trains [5,6,7,8,9]. To keep up safety standards, new halogen-free flame retardants are required [8]. Substitutes, such as inorganic compounds, for example, polyphosphates or organic phosphates, or nitrogen compounds, have been developed [10] with a focus on nitrogen and phosphor-containing chemicals due to a P-N synergistic flame retardant effect [11,12,13,14,15]. A drawback, however, is the low stability with respect to washing and mechanical abrasion, [16,17] which limits the applicability of such flame retardants for textile finishing. Other studied approaches for flame retardant textiles include layer-by-layer coating of, for example, cationic polyelectrolytes, such as polyallylamine, [18,19,20] chitosan [21,22,23,24] or polyethylenimine [25,26], with anionic nanoclays [18,24,26,27,28], polyphosphates [18,21,25,29] or DNA [30]. Further investigations concentrated on sol–gel chemistry [31,32,33,34,35,36], carbon nanotubes [29,37,38], polycarboxylic acids [39,40,41,42], protein-based coatings [43] and photo-[44,45,46] or plasma-grafting [27,47,48]. An overview of phosphorous based flame retardants is given by Salmeia et al. [49]. Further information can be found in many reviews [49,50,51,52] and books [2,9,16,53].

Due to their high limiting oxygen index (<60) [54,55] phosphazenes are promising flame retardants. Although their utility was proven with good flame retardant properties in polymer blends, [56,57,58,59,60] very few results have been reported for their use for textile finishing. Shukal and Arya [61] showed that poly(fluorophosphazenes) in combination with nowadays undesired organo-bromine compounds improve flame retardant properties of PET. The individual contribution of bromine-compounds and polyphosphazene to the total effect remains unclear. Other examples have been described in various patents, where the phosphazene derivatives are used for textile finishing [62,63] or fiber polymer blending [64,65,66,67] or in a polyurea coating for textiles [67]. Up to now, no commercial textile finishing agent based on phosphazenes is available. The reason for this is the lack of existing phosphazenes with anchor groups for the durable fixation on the fiber surface and an appropriate process technology. Motivated by this situation and our experience in the application of photo-grafting for textile finishing [44,68,69,70], we developed a flame retardant based on allyloxy-polyphosphazenes, which covalently bound to cotton and cotton/polyester blends using photochemical reaction conditions [71]. This finish displayed good durability and good flame retardant properties, but industrial application failed due to the lack of water solubility of the allyloxy-polyphosphazenes. Wang et al. [72] combined a cyclophosphazene with an aminosilane and needed an add-on of 19% on the cotton to obtain and limiting oxygen index (LOI) of 24.5%. On the other hand, three studies about cyclotriphosphazenes bearing an allylamino group were published, in which the authors demonstrated the photo-curing [73] of allylamino cyclotriphosphazene and its applicability as a flame retardant for cotton fabrics [74] with a 30 wt% add-on and further, for polyester resin [75]. In terms of synthetic routes, cyclotriphosphazene derivatives are highly interesting as they can be readily synthesized in high yields in comparison to polyphosphazenes, which need one more polymerization step with low yield and have a high energy consumption. However, finishing of different textiles with cyclotriphosphazenes has not been studied in great detail. For this reason, an allylamino cyclotriphosphazenes (CPZ) was synthesized and tested with regard to its efficiencies as a flame retardant for CO, PET, PA, and CO/PET- or COPA-blended textiles in the presented study. Inspired by the work of Qian et al. [76,77] who demonstrated thiol-ene reactions in water with allylamine and propargylamine polyphosphazenes, water-soluble cyclotriphosphazene derivatives were thus adapted for textile finishing. Up to now, no work has been reported on the utilization of CPZ as a green flame retardant in water-based finishing protocols of textiles. Our previous work established polyphosphazenes as a solvent-based flame retardant finishing for textiles [71]. Based on this work, a water-based finishing for textiles based on a reactive CPZ was developed.

## 2. Results and Discussion

### 2.1. Synthesis and Characterization of CPZ

Hexa (allylamino) cyclotriphosphaze was synthesized following the procedure of Ahn et al. [73] (Scheme 1). The NMR data were in full agreement with those reported. To obtain a water-based solution of CPZ, the method of Qian et al. [76,77] for allylamino-polyphosphazenes was adopted, which proved that the phosphazenes remained stable under acid conditions. Therefore, the solubility of the CPZ was tested in different aqueous acids (H_3_PO_4_, H_2_SO_4_, HCl). In comparison to the work of Qian et al., a good solubility not only using phosphoric acid but also in hydrochloric acid and in sulfuric acid was found. Depending on the pH value of the acid solution, the solubility of the CPZ changed. The solubility in the range of 0.1 to 1 M of the acids with 1 to 30 wt% of CPZ was tested. The solubility of CPZ in aqueous acid solutions is obtained by protonation of NH-groups. Table 1 shows the solubility of the CPZ at different acid concentrations. For instance, a concentration of 15 wt% CPZ is only soluble in water at an acid concentration of 0.5 mol/L.

To understand the thermal stability of the CPZ, differential scanning calorimetry (DSC) and thermogravimetric analysis (TGA) measurements of photo crosslinked CPZ were conducted, results of which are shown in Figure 1. The DSC curve shows that the crosslinked CPZ lost the incorporated water, followed by a complex endothermic decomposition starting at 150 °C. The TGA curve of the crosslinked CPZ displays a two-step weight loss with ~50% residual mass (char) after heating at 700 °C while the DTG thermogram shows two maxima for each. Allcock et al. [78,79,80] described that these reactions included decomposition of the side-chains, different thermal rearrangement reactions, and pyrolytic cross-linking. The opening of the CPZ rings was also involved in the decomposition reactions at a temperature higher than 250 °C.

### 2.2. Textile Finishing and Characterization

For the evaluation of a finish based on the synthesized reactive flame retardant, different kinds of fabrics were chosen, which are commonly used for textile applications with a high demand for flame retardant properties (e.g., protective clothing, composites, car interiors, furniture). These fabrics include: one cotton fabric and two PET/CO blends and one COPA 6.6 blend (also called NyCO); all blended fabrics are 50/50 mixtures. Additionally, plain PA 6.6 and PET fabrics were also tested. PET/CO 1 was a warp sateen weave fabric with a CO and a PET side. Fabrics PET/CO 2 and COPA were manufactured from core-shell yarns with a PET and PA core, respectively. The final textile finishing was achieved by using the HCl solution of the CPZ. The loading with the flame retardant CPZ, also termed the add-on value, was controlled exclusively by the concentration of the CPZ solution, while the same amount of CPZ solution was applied to the samples throughout. Since the studied textiles each had a different fabric weight per unit area, we tried to finish all fabrics with the same amount of CPZ according to the area of the fabrics. This means that the obtained flame retardant loadings differ as the fabrics had different area weights. Five different concentration were used for the finishing of six different kinds of textile.

It should be noted here, that passing a certain standardized flame test strongly depends on the flame retardant loading. On the other hand, the allowable amount of the deposited flame retardant is actually limited to a certain degree by the desired application. For some textile applications, such as personal protective equipment (PPE), the haptic properties should ideally stay unchanged. Thus, a high add-on resulting in a change of the haptic properties of the textile is unfavorable.

After the application of the finishing solution, the textiles were irradiated with a broad-band UV-lamp in the presence of a photosensitizer. To remove all unbound CPZ and remaining acid, the textiles were washed before further analyses and all flame tests. In Table A1 (Appendix A) the CPZ concentrations and pH-values of the used HCl solutions and the obtained add-ons before and after washing are listed. The samples and conditions for analysis and external flame test are marked. For TGA, ATR-IR, and microscale combustion calorimetry (MCC) studies, samples with about 10 wt% add-on were used. In the case of PET/CO1, a double coating was necessary.

Figure 2a shows the add-on value measured gravimetrically as a function of the applied amount of gram CPZ per square meter of textile and the used concentration of CPZ after one washing cycle (for the unwashed samples see Figure A1 in the Appendix A). It can clearly be observed that the add-on value increased with an increasing amount of offered CPZ. As stated above, the different add-ons were observed on the different fabrics at a fixed FR. PET/CO 1 displayed the lowest add-on value, while both PET/CO 2 and PA fabrics showed the highest. This depended on the different area weight (PET/CO 1: 340 g/m^2^, PET/CO 2: 170 g/m^2^) and suction properties of the fabric.

From an application point of view, washing resistance is essential. Therefore, the modified textiles with a coated with 3 g/m^2^ CPZ or 15 g/L were washed in a Linitester according to EN ISO 105-C06. The washing stability measured after one, three, and six washing cycles is shown in Figure 1. After the first washing cycle, a weight loss of the grafted CPZ for all textiles was observed, while the add-on values for all fabrics stayed constant for the next washing cycles resulting in a stable finishing of CPZ.

The SEM images of the treated fabrics after one washing cycle show that a smooth CPZ film coating uniformly dispersed on the surface and around the fibers. It can also clearly be observed that the treated CO and PA fabrics almost retain their original structure, indicating the infiltration of CPZ into the inner fibers rather than coating only onto the surface of those fabrics (Figure 3).

### 2.3. Thermal Stability

The thermal stability and the flame retardant mechanism of CPZ were investigated using TGA and microscale combustion calorimetry (MCC). With the TGA measurement, the thermal behavior of the CPZ in the solid phase during the char formation can be assessed, while using MCC information about the behavior of the material in the gas phase can be obtained by measuring the heat release. Lyon et al. [81] reported that the heat release capacity (HRC) measured by MCC can be correlated with the flame retardant properties, for example, the UL 94 flame test. Important indicators for a good flame retardant are the reduction of both the heat release rate and heat release capacity of the modified fabrics upon combustion.

#### 2.3.1. Thermal Properties

The thermal stability was measured by TGA with a heating rate of 20 K/min under nitrogen. The results are listed in Table 2. The add-on value for all measured samples was ~10 wt%. Many authors reported that good flame retardant properties of finishing textiles correspond with a lower decomposition temperature, which is determined at a 5% weight loss (T_–5%_), and an increase of the remaining residue or char (Res) compared to the unfinished textile. Figure 4 shows the (derivative) thermograms of both finished and unfinished fabrics. Around 100 °C, about 1% of adsorbed water evaporated. Cotton fabric showed one single thermal decomposition step at 380 °C, which is related to the depolymerization by trans-glycosylation reactions of cotton [32]. In contrast, the decomposition reactions of PA and PET were found to take place at higher temperatures ~450 °C. As expected, the blended textiles exhibited two independent decomposition steps, where the first represents the depolymerization of CO at 380 °C and the second is attributed to PET or PA, respectively (~450 °C).

For the CPZ finished textiles an earlier decomposition (T_−5%_) in comparison to the untreated materials was observed (see Table 2). In the case of CPZ-modified CO (Figure 5a), the decomposition started at ~300 °C and reached a maximum at a temperature between 310 °C and 320 °C with a higher residual mass at 700 °C compared to the pristine CO, which decomposed faster with higher mass loss. This increase in the char formation is typically related to the effect of the flame retardant, which forms phosphoric acid upon thermal decomposition and which is capable of inducing a catalytic water elimination leading to a graphitization of cotton [32,82].

For PET (Figure 5b) and PA (Figure 5f), two steps of decomposition in the presence of the CPZ were observed, whereby the first step exhibited a decomposition at about 380 °C and 450 °C for PET and PA, respectively. The second decomposition step of PET at 450 °C had a higher weight loss (~40%), while the higher weight loss of PA (~60%) was observed in the first decomposition step. The first decomposition reactions could be induced by the decomposition of the CPZ, which had a decomposition step at 391 °C. The decomposing CPZ seemed to react with the (decomposing) PET and PA, changing both mass losses and residual masses for the corresponding compounds, whereby the weight losses in this decomposition step indicated that PA had a higher reactivity than PET. For PET, these reactions have no significant influence on the total amount of decomposed fabric as the residue formed is just 1% higher than the unmodified PET, while for PA, ~11% residue is obtained. It should be noted that low weight loss of about 2% was also observed for CPZ@PET at a temperature of 230 °C. At this temperature, the CPZ itself showed a decomposition reaction, as seen in Figure 1. Therefore, it may be assumed that this initial decomposition reaction has no influence on the decomposition of PET. However, based on the assumption that a good textile flame retardant gives a higher residual mass, it can be concluded that the CPZ is not an effective flame retardant for PET.

The CPZ-coated blend fabrics exhibited an individual thermal decomposition of the relative component (Figure 5c–e). Regarding the PET/CO blends (both 1 and 2), two decomposition steps were observed with a shift in the temperature of the cotton decomposition as was already observed for the CO fabric. An increase in the char formation of about 18% and 24% was also obtained for PET/CO 1 and PET/CO 2, respectively. This difference in the mass loss may be related to the differences in the textile fabrication of both blends; CO/PET 2 is a core-shell fabric, where the PET core is shielded to a certain degree by the decomposing cotton fibers. For CPZ@COPA, three decomposition steps were observed (Figure 5e), where the PA component displayed two thermal decompositions. The CO decomposition in the blend fabric was not influenced by PA.

From the MCC measurement, the dependence of the thermal decomposition of the fabrics on the energy of the decomposed products in the gas phase was obtained. Although the heating rate of the MCC measurement (1 K/s) is higher in comparison to TGA (0.3 K/s), similar observations regarding the decomposition steps, as well as the decomposition temperatures, were obtained from MCC measurements (Figure 6). As observed in the TGA, pristine CO, PET, and PA showed just one heat release peak indicating one decomposition reaction, while the blends showed two peaks belonging to each fiber polymer. In addition, in the presence of CPZ, similar decompositions were observed with a broad peak of the CO component. All CPZ-finished textiles showed a reduced HRC and THR with a higher char yield. For CO, PET/CO 1 and 2, and COPA the HRC was <160 J g^−1^ K^−1^, which means that the material was self-extinguishing and could obtain a UL 94-V0 rating as suggested by Lyon et al. [81]. However, the precise correlation of MCC-parameters with the textile standard flame retardant test is not known. Figure 5 displays char yield, HRC, and THR measured by MCC. Table A2 (Appendix A) displays the peak of heat release, the peak of the heat release rate, and the peak of temperatures. As is clearly observed, the HRC, as well as the THR (Figure 6a,b), decreased for all coated fabrics indicating that CPZ decelerated a developing fire by the promotion of char formation (Figure 6c).

#### 2.3.2. Flame Test

Depending on the desired application, different flame retardant tests are requested. Therefore, the flammability was examined by four different standard textile tests, which are summarized in Table 3. The limiting oxygen index value (LOI) provides information about the oxygen amount required to keep a sample burning for 180 s. High LOI values indicate a high required oxygen concentration, and therefore, the sample is harder to burn/oxidize. DIN 75,200 is a test which is relevant to the automotive section, DIN EN ISO 15,025 is required for protective clothing (PPE), and EN ISO 11925-1 is required for building materials in public buildings including textiles. With the changing sample orientation from horizontal to vertical and changing the flame application from the face of the sample to the bottom edge, the difficulty of these tests increases, because the position of the tested material is more and more orientated in the direction of the heat flux. For the first evaluation, we followed the DIN EN 15,052 flame test with a reduced sample size of 50 × 100 mm to require less CPZ for the coatings. Our experience showed that we obtained to standard comparable results [71].

To measure the flame retardant properties of CPZ-modified textiles, samples coated with different concentrations of CPZ solution (Figure 1) were tested according to DIN EN ISO 15025, whereby smaller samples sizes of 5 × 10 cm were used. In Figure 7, the time of burning after flame application (after flame time (*af*)) and the add-on is shown (Table A3 of the supporting materials gives the values). Only for PET/CO 2 was an after glow observed. For CO, PET/CO 1 and 2, and COPA, it can be observed that by increasing the add-on, the *af* dropped to zero, and the fabrics passed the test. These four fabrics have in common an already a low amount of the CPZ (5–10% add-on), which reduced the after-flame time to 0. For PA and PET, an increase of the after-flame time was observed. This can be explained by the fact that finishing with a flame retardant changes the melting behavior in that the coating prevents the polymer melting from the flame. The prevention/changing of the melting behaviors accordingly leads to an increase in flammability of CPZ@PA and CPZ@PET with add-ons of up to 10%. With a higher add-on, the after-flame time dropped again, because the CPZ finish works as a flame retardant as was expected by the reduced HRC. The same flame test was also used to evaluate the washing resistance. The flame test for finishing solutions with 30 g/m^2^ after one, three, and six and for 40 g/m^2^ after six washing cycles was evaluated. The flame test after washing is shown in Figure 8. The flame retardant works even after six washing cycle for CO and the blended fabrics.

The results obtained from the flame test, according to DIN EN ISO 15,025, were used to select samples for the flame test following other standard tests. Due to the fact that the CPZ only showed a promising flame retarding effect for CO and the blend fabrics, only these materials were examined in the following. To represent different application areas LOI, DIN 75,200 (automotive), DIN 15,025 protective clothes (PSA), and DIN 4102-1-B2 building materials were chosen as standard measurements. The add-on for LOI and DIN 75,200 was in the range of 5% to 6 %. For DIN EN ISO 15,025 and DIN 4102-1-B2 an add-on between 9% and 11 % was used. The results for all test are given in Table 4.

Results may be summarized as follows: The LOI measurement showed a significant increase in all cases. All fabrics finished with CPZ passed the DIN 75,200 test for automotive textiles, which specifies a burning speed below 120 mm/min. The PSA-test was passed, while the challenging DIN 4102-1-B2 standard for building materials indicated an improvement of flame retardant properties, but the standard was not passed by any of the samples. It is assumed that the decisive factor is the edge impingement performed in this test in contrast to the other testing methods applied.

#### 2.3.3. Char Analysis

By analyzing the residual char of the fabrics with 10wt% add-on after the flame test, according to DIN EN ISO 15,025, some additional conclusions on the flame retardant mechanism could be obtained. IR spectra of CPZ finished fabrics exhibited a characteristic peak at ~1450 and ~1100 cm^–1^ assigned to the P-N-C and P-N-P-bond (cf. also Figure 9 and Figure 10a,b). The broad absorption bands in the range of 3000 to 3600 cm^–1^ were due to the H–N stretching, which belongs to both CPZ and amide groups of PA. For PET and PA samples, the IR-spectra of the charred area were not changed in comparison to the original material, while charred CO showed a significant change in the IR-spectrum (Figure 9a,c). The behavior of PET and PA is explainable by the fact that both materials melt away. For the CPZ@CO, the IR spectrum of the char differed strongly from the unfinished fabric (Figure 9b,d). For the CPZ@PET, the charred IR-spectrum was similar to PET (Figure 9c,d). For the PET/CO 1 and 2, the IR spectra of the char of the untreated and treated fabrics were similar to each other (Figure 9c,d) and similar to the unburnt PET (Figure 9a). This behavior has also been observed for textiles finished with a polyphosphazene [71]. This finding can be explained by the melting of the PET. After the flame was removed, the PET was still liquid and flowed to the surface of the textile.

The same observation can be made for the PA. The IR spectrum of the PA char was similar to the spectrum of the unburned PA. The char of CPZ@PA showed IR signals of PA (between 1500 and 4000 cm^−1^) and new IR-signals comparable with charred CPZ@CO (between 600 and 1250 cm^−1^).

The relevant IR-spectra of the blend fabrics are summarized in Figure 10. Significant differences were found in the char of untreated and finished COPA fabric (Figure 10c,d). The char spectrum of the untreated sample showed a mixture of the CO char product (peaks between 600 and 1250 cm^−1^) and a contribution of PA (band between 1500 and 4000 cm^−1^) (cf. Figure 9c and Figure 10c for the CO and PA spectra). In case of the burnt CPZ@COPA fabric, a similarity to the char of CPZ@CO (cf. Figure 9d and Figure 10d) was found. The important signals of charred CPZ@CO, CPZ@COPA, and CPZ@PA were around 1600 cm^−1^, which can be assigned to a C=C absorption, and the broad signals at 1200 cm^−1^, which can be assigned to different groups, such as P=O, P-O-P (symmetrical stretching,) and P=N. The signal at 900 cm^−1^ belongs to P-O-P (symmetrical stretching). Based on these observations, we can conclude that in the presence of CPZ charring reactions of both PA and CO can be induced. The char spectrum of PET/CO 1 and 2 @CPZ was similar to the char of the unfinished blends. In both cases, the char spectra showed the presence of PET. This behavior has also been observed for textiles finished with a polyphosphazene [71]. This finding can be explained by the melting of the PET. After the flame was removed, the PET was still liquid and flowed to the surface of the textile. The decomposition of PET remained mainly unchanged in the solid state. The charring reaction with CO and PA led to the formation of phosphoroxinitrids (1200 and 900 cm^−1^), a glass-like compound, which formed a layer passivating the fire. Due to the melting of the PET, the underlying of the phosphoroxynitride was not detectable.

The char morphology was studied by SEM. The SEM images of the treated fabrics show a smooth CPZ film coating uniformly dispersed on the surface and around the fibers. The finished fabrics almost retain their original structure, indicating the infiltration of CPZ into the inner fibers. (Figure 11) After burning, all fabrics showed a different morphology compared to the treated fabrics before burning, but almost similar behavior could be observed in the burning area. All fabrics that contained CO showed a char formation resulting from the reaction of CPZ with cellulose, while the PET fabric formed a bulkier rougher film compared with that of untreated fiber, which may be related to the melting of PET. On PA, a rougher surface was observed. For the different blends, a film formation on the surface with a formation of gas bubbles which were released during the combustion was observed. This bubble formation was verified by comparing the cross-sections of the charred areas (Figure 11). In the case of CPZ@CO, the typical formation of hollow fibers was observed, and this behavior was also conserved in the CPZ@blend fabrics (PET/CO1, PET/CO2, and COPA). This hollow fiber formation depended on the charring reaction, whereby the protective glassy layer forms the hollow fibers. The observed higher roughness of CPZ@PA also originated in foaming. The char of CPZ@PET was too brittle for a cross-section measurement. The film formation in case the PET/CO blends correlated with the IR observation, where mainly PET was observed on the surface.

## 3. Materials and Methods

### 3.1. Textiles and Chemicals

The experiments were carried out using a commercial cotton woven fabric (twill 3/1, 230 g/m^2^, white, CHT R. Beitlich GmbH, Tübingen, Germany), a commercial 50/50 CO/PET woven fabric (warp sateen 4/1, 340 g/m^2^, orange, Huntsman Textile Effects GmbH, Langweid am Lech, Germany (PET/CO1)), a commercial 50/50 CO/PET core yarn fabric (twill 2/1, core PET, CO shell, 170 g/m^2^, camouflage, Bluecher GmbH, Erkrath, Bluecher GmbH, Germany d(PET/CO1)), and a 50/50 COPA core yarn fabric (twill 2/1, core PA, CO shell, 190 g/m^2^, camouflage, Bluecher GmbH, Erkrath, Germany). Hexachlorotriphosphazene was obtained from Eurolabs Limited (Crawley, UK) and allyl amine (≥99%) from Merck (Germany). Tetrahydrofuran (THF, ≥99.9%) were obtained from Carl Roth (Karlsruhe, Germany). Triethylamine (Et_3_N, ≥99%), was obtained from Sigma-Aldrich (St. Louis, USA). All solvents were dried before use.

### 3.2. Instrumentation

Fourier transform infrared spectroscopy (FT-IR) was carried out using an IR Prestige by Shimadzu (Europe); for the attenuated total reflection (ATR) mode, a Golden Gate, diamond crystal, Specac (UK) with a resolution of 4 cm^−1^ was used. Differential scanning calorimetry was measured using a DSC Q20 (TA Instruments, New Castle, USA) under 50 mL/min N_2_ and a heating rate of 20 K/min. Thermogravimetric analyses (TGA) were performed with a Discovery TGA55 (TA Instruments, New Castle, USA) under 50 mL/min N_2_ and a heating rate of 20 K/min in aluminum oxide crucibles and with a Mettler Toledo TGA/SDTA 851 in N_2_ (gas flow rate of 30 mL/min) and a heating rate of 10 K/min in aluminum oxide crucibles. Microscale combustion calorimetry (MCC) were measured with FAA Micro Calorimeter (FTT Fire Testing Technology, West Sussex, UK) following ASTM. ^1^H and ^31^P nuclear magnetic resonance spectroscopy (NMR) was conducted using a Bruker DMX300 (USA) and a deuterated solvent as internal standard. Scanning electron microscopy (SEM) was done using an SEM S-3400 N II, Hitachi High-Technologies Europe. An Ultraviolet A (UVA) print system lamp (Type 100–200, HPV-E2, H emitter with dichroitic reflector for IR reduction, power 200 W/cm, Hoenle UV Technology, Gräfling, Germany) served as a broadband UV-source. Standardized washing tests were carried out using a linitester by Atlas Material Testing Technology (Rock Hill, USA).

### 3.3. Synthesis and Characterization of Hexa (allylamino) Cyclotriphosphazene (CPZ)

Freshly sublimated hexachlorotriphosphazene (20 g, 58 mmol) was solved in 200 mL dry THF under argon atmosphere. A mixture of 8 eq. (63.1 g, 700 mmol) triethylamine and 8 eq. allylamine was solved in 100 mL THF and dripped into the hexachlorotriphosphazene solution over 2 h and refluxed for further 2 h. Afterward, the reaction mixture was poured into 300 mL of water, and the product was extracted with CHCl_3_. After evaporation of the solvent, a yellow solid with a yield of 92% was obtained.

^1^H NMR (300 MHz, CDCl_3_): δ 3.04 (s, 1 H), 3.38 (s, 2 H), 5.81 (ddt, J=17.2, 10.5, 5.4 Hz, 2 H) 4.91 (dq, J=17.2, 1.8 Hz, 1 H) ppm. ^31^P NMR (122 MHz, CDCl_3_): δ 18,96 (s) ppm. IR (ATR): 783 (P-N-C), 848, 912, 995, 1005 (P-N-backbone), 1080 (P-O-R), 1170 (P=N-backbone), 1415 (P-N-C), 1647 (C=C), 2951 (C-H, sp^3^), 2980 (C-H, sp^3^), 3015 (C-H, sp^2^), 3066 (C-H, sp^2^), 3170 (N-H) cm^−1^.

### 3.4. UV Treatment and Characterization of the Modified Fabrics

The textiles were immersed in the CPZ solution. For a flame test sample of 5 x 10 cm (0.005 m^2^), 1 mL of the hydrochloric aqueous CPZ solution with 0.4% Irgarcure 2959 (see supporting information Table A1 for all concentration and add-on values) was used. For bigger samples the amount of CPZ solution was scaled, for 1 m^2^ of textile 0.2 L of CPZ-solution was needed. Samples with about 10 wt% CPZ add-on after washing of PET/CO1 were obtained by double coating with 20% CPZ-solution in 1 mol/L HCl (PET/CO1 9.6 wt%). The immersed samples were irradiated for 10 min in an argon atmosphere. The distance between the light source and the sample was 20 cm. Subsequently, each sample was washed once in a textile linitester to remove non-bonded CPZ; afterwards the textiles were dried at RT and weighed. The add-on (A [wt%]) was calculated using equation 1, with *W*_0_ = before irradiation, *W*_1_ = after irradiation and washing.

(1)$$A=\frac{W_{1}-W_{0}}{W_{0}}\times 100\%.$$

### 3.5. Measuring of the Thermostability

TGA was carried out under nitrogen (50 mL/min) with a heating rate of 20 K/min in an aluminum oxide crucible with precisely weighted sample sizes between 7 and 20 mg.

MCC measurements were done following ASTM D 7309 Method A. Sample weights were selected between 15 and 30 mg to obtain an O_2_ concentration drop of 10%. Samples were pyrolyzed under N_2_ with a heating rate of 1 K/s up to 750 °C and a combustor temperature of 900 °C. The data were analyzed by Microcal Origin 2018b Professional.

### 3.6. Washing Resistance

To evaluate the washing fastness of the modified textiles, the materials were washed up to six times in a linitester according to EN ISO 105-C06 (liquor volume 150 mL, liquor ratio 1:80, ECE detergent 4.0 g/L, 30 min, 40 °C). The samples were then dried at room temperature and weighed.

### 3.7. Evaluation of the Flame Retardant Properties

All flame tests and LOI measurements were carried out after at least one washing cycle. The flame retardant properties were measured according to DIN EN ISO 15,025 (protective clothing—protection against heat and flame—method of test for limited flame spread), with a reduced sample size of 5 cm × 10 cm and a Proxxon lighter (combustion gas butane). In addition, selected samples were tested externally by Staatliches Pruefamt fuer das Textilgewerbe, University of Applied Science Hof (Germany). One sample set was tested according to DIN EN ISO 15025, two sets according to DIN 75,200, respectively, US FMVSS 302 or ISO 3795 (motor vehicle safety standard), five sets according to EN ISO 11925-2 German “Kleinbrenner“ method and five sets according to DIN EN ISO 4589-2 (limiting oxygen index (LOI, 0.5% steps)).

## 4. Conclusions

Different phosphazenes are proposed as flame-retardant materials in literature, but the investigation of such compounds as broadly applicable flame retardants for textile finishing is scarce to date. With this paper, we demonstrated the usefulness of a cyclotriphosphazene derivative as a flame retardant for textile finishing. The reported CPZ can be processed from water and by UV-cross linking, whereby a washing resistant coating is obtained. Thus, two important conditions for industrial application—finishing from aqueous systems and durability—are met. A further important advantage of the CPZ is that only a single step synthesis is required to obtain the product. The used CPZ is an effective flame retardant for CO and CO blends with PA and PET. For pure PA and PET fabrics, a reduced HRC was observed, but flammability still occurred. Depending on the application, an add-on between 5% and 10% was necessary to obtain the required flame-retardancy.

With regard to the mechanistic aspect of the CPZ, two potential mechanisms of action are proposed. The first is the formation of an acid, which catalyzes graphite formation for cellulosic materials and an acid-catalyzed decomposition for PA. It, furthermore, improves char formation of these materials. The second mechanism is the formation of a glassy insulating layer of phosphoroxynitrides. For a better understanding of the complete flame retardant mechanism of the presented phosphazene-based flame retardant, in-depth mechanistic studies are currently in progress, which include the comparison between different types of poly- and cyclophosphazens.

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
