# Peer review of "A Green Water-Soluble Cyclophosphazene as a Flame Retardant Finish for Textiles"

_molecules, 2019, doi:10.3390/molecules24173100_

Round 1

Reviewer 1 Report

This is an interesting manuscript. However, it requires some corrections.

In fact, hexa(allylamino)cyclotriphosphazene (CPZ) was used as flame retardant for modification of different textile materials, in a form of salts: phosphate, sulphate or hydrochloride, which were soluble in water.

What does it mean "polymer" in Fig. 1b and Fig. A1 ? Does it mean CPZ salt ? Which one was used: phosphate or sulphate, or hydrochloride ? It should be explained and corrected.

What were concentrations of CPZ salts and pH of their water solutions in each case ? This comment concerns all presented results, including ATR-IR spectra and TGA data.

In order to avoid a presence of acids on textiles after their modification with acidic solutions of CPZ salts, I recommend their isolation and dissolution in water before soaking textiles.

Which absorption peak, in FT-IR spectra, corresponds to protonated N-H group ?

A LOI value should reach 28 or even 32 for some applications of textile materials, e.g. in common transportation or in public buildings.

A statement "... water soluble and permanent systems are missing" on page 1 (in line 15) of an Abstract is not grammatical and it should be corrected.

Author Response

Dear reviewers we thank you all for the useful comments.

All your comments about failures in the manuscript have been considered and corrected. To improve the quality of the publication we did a careful prove reading.

Reviewer 1.

What does it mean "polymer" in Fig. 1b and Fig. A1 ? Does it mean CPZ salt ?

We corrected Fig.1 and A1 to CPZ. 

Which one was used: phosphate or sulphate, or hydrochloride ? It should be explained and corrected. What were concentrations of CPZ salts and pH of their water solutions in each case? This comment concerns all presented results, including ATR-IR spectra and TGA data.

We extended the manuscript for the missing information. All finishing were done with a hydrochloric CPZ solution. We added Tab. A.1 were the used concentrations for the finishing is given with the obtained add-ons before and after washing. This table also contains the information about the used conditions to produce the fabris for the different analysis and external flame tests. There is one point, where I’ve to say that forgot an information. The 10 % add-on for PET/CO1 where obtained by a double coating.

To make the experimental section better repeatable we extend it. For TGA, ATR-IR and MCC we used an add-on of about 10 %.

Reviewer question about pH:

We had not been interested into the pH-value of the CPZ-finishing solution, therefore we did not measured it. We just know the pH-value of the used HCl, which we have included into the paper. The pH-value of the finishing solution should be similar to the acid.

In order to avoid a presence of acids on textiles after their modification with acidic solutions of CPZ salts, I recommend their isolation and dissolution in water before soaking textiles.

We already tried this and observed a low solubility of the CPZ. The ammonium-group is a strong acid, and in water it dissociates and the free CPZ is formed. The unprotonated CPZ is not water soluble.

Which absorption peak, in FT-IR spectra, corresponds to protonated N-H group ?

We have clarified the manuscript, samples (TGA, IR, MCC) were measured after washing. As explained before, the amine is deprotonated in water, and therefore no R2NH2+ is found.

A LOI value should reach 28 or even 32 for some applications of textile materials, e.g. in common transportation or in public buildings.

I checked the new nominate standards “Fire classification of construction products and building elements“ EN 13501-1:2007 and the former DIN 4102-B2 both does not contain any LOI values. We also checked DIN 5510-2 “Preventive fire protection in railway vehicles” an LOI values isn’t requested there. Can you give a reference, where we can find these LOI values? Then we can include them into the discussion. Thanks.

Reviewer 2 Report

The authors in their manuscript addressed an interesting topic. The manuscript is suitable for publication after some minor corrections.

1. A related review article that published in polymers )Polymers 2016, 8(9), 319; https://doi.org/10.3390/polym8090319) is not mentioned by the authors. This review article summarizes some data which is related to the flame retardant additive that used in this manuscript.

 2. A typo is found in the title of fig. 3e.

 3. Many sentences are short and incomplete which may confuse the readers. Ex. :

a) Line 207-208: this sentence is unclear.  Figure 3b and table 3 show a shift in the decomposition temperature. Can the author comment on this.

b) same in line 223-224. The sentence is unclear

 4. The orders of figures and tables is confusing.  i.e. the figures and table are far from where they mentioned for the first time.

 5. it is recommended if figure 5 can be changed to a table.

Author Response

Dear reviewers we thank you all for the useful comments.

All your comments about failures in the manuscript have been considered and corrected. To improve the quality of the publication we did a careful prove reading.

Reviewer 2

A related review article that published in polymers )Polymers 2016, 8(9), 319; https://doi.org/10.3390/polym8090319) is not mentioned by the authors. This review article summarizes some data which is related to the flame retardant additive that used in this manuscript.

Thank for the information about this publication We include it into the general part ofr the introduction and into the discussion about phosphazens as flame retardants. After reading the review we decided to extend this part with the missing work of Wang et al.

“Wang et al. combinend a cyclophosphazen with an aminosilane and needed an add-on of 19 % to obtain and LOI of 24.5 %. However, finishing of different textiles with cyclotriphosphazenes is not studied in detail.“

Many sentences are short and incomplete which may confuse the readers.

While the prof-reading we checked all sentences carefully, and change sentences with this failure to make them more understandable.

The orders of figures and tables is confusing.  i.e. the figures and table are far from where they mentioned for the first time.

We restructured the manuscript and moved the figures in better postisiton.

it is recommended if figure 5 can be changed to a table.

This topic was also discussed by use before the publication. We decided to use the graphic to give the reader have an easier comparisons of the observed changes of the MCC results. I know commonly a table is used to present this results, therefore, we have included the table in the supporting materials (Tab. A2).

Reviewer 3 Report

This manuscript reports the development of hexa(allylamino)cyclotriphosphazene which can be attached by photo-grafting to provide a durable flame-retardant finish for cotton and cotton blend fabrics. This represents something of an advance in that the initial coating of the grafting agent on the fabric can be done in aqueous solution.

The manuscript will need revision to improve clarity/accuracy and readability. Corrections are pencilled-in directly on pages of the manuscript attached. These are indicative of the kinds of changes needed throughout. A Table is not active - it cannot show - the data contained in the Table show. Reference to "synergism" should be omitted. For the reference cited, the impact of two agents is not even additive. Such phrases as "in the literature" should be omitted. Author's names and personal pronouns should be omitted.

Author Response

Dear reviewers we thank you all for the useful comments.

All your comments about failures in the manuscript have been considered and corrected. To improve the quality of the publication we did a careful prove reading.

Reviewer 3

A Table is not active - it cannot show - the data contained in the Table show.

Dear reviewer, pityly you did not name the Table, and I can’t find the failure. I checked the document and all tables are working fine.

Reference to "synergism" should be omitted. For the reference cited, the impact of two agents is not even additive.

Synergism of P-N for flame retardant applications is a proven factor. We included some more references which all prove P-N-synergism.

Such phrases as "in the literature" should be omitted.

We have taken this phrase out of the manusricpt.

Author's names and personal pronouns should be omitted.

Here I don’t not agree with the reviewer. The usage of authors names in manuscript is personal style. For the usage of pronouns or not, it is also a question of personal style. We check the manuscript and removed them in some cases.